# Investigating Online Community Engagement through Stancetaking

**Jai Aggarwal**  **Brian Diep**  **Julia Watson**  **Suzanne Stevenson**

Department of Computer Science
University of Toronto
{jai, bdiep, jwatson, suzanne}@cs.toronto.edu

## Abstract

Much work has explored lexical and semantic variation in online communities, and drawn connections to community identity and user engagement patterns. Communities also express identity through the sociolinguistic concept of stancetaking. Large-scale computational work on stancetaking has explored community similarities in their preferences for stance markers – words that serve to indicate aspects of a speaker's stance – without considering the stance-relevant properties of the contexts in which stance markers are used. We propose representations of stance contexts for 1798 Reddit communities and show how they capture community identity patterns distinct from textual or marker similarity measures. We also relate our stance context representations to broader inter- and intra-community engagement patterns, including cross-community posting patterns and social network properties of communities. Our findings highlight the strengths of using rich properties of stance as a way of revealing community identity and engagement patterns in online multi-community spaces.

## 1 Introduction

Communities vary in their language choices, often in ways that are indicative of the community's shared interests, values, and norms – their *community identity*. Sociolinguists refer to such spaces as *communities of practice* (Eckert and McConnell-Ginet, 1992; Holmes and Meyerhoff, 1999), and have explored how they vary in their in-person linguistic practices (Cheshire, 1982; Eckert, 2000). Computational work has explored variations of word and sense usage in *online* communities of practice (e.g., Zhang et al., 2017; Del Tredici et al., 2018; Lucy and Bamman, 2021), highlighting relationships between linguistic and non-linguistic aspects of community identity, including user retention, community size, and network structure.

Community identity is also expressed through **stancetaking** (Bucholtz and Hall, 2005), in which speakers position themselves in various ways – affectively, evaluatively, epistemically, etc. – relative to a topic or a conversational partner (e.g., Du Bois, 2007). Sociolinguistic work (e.g., Kiesling, 2004; Bucholtz, 2009) has studied how communities vary in their use of **stance markers**. Stance markers are words that demarcate stance, including, among others, intensifiers (*really*, *insanely*, *terribly*), modals (*might*, *should*), and evaluative words (*like*, *love*, *hate*). For instance, compare these sentences from three Reddit communities:

1. *I hope you are **insanely** proud of yourself!*
   [r/pornfree, helping people overcome porn addiction]

2. *I hope you are **incredibly** proud of yourself!*
   [r/progresspics, support for sharing fitness progress]

3. *Both of you are **incredibly** stupid*
   [r/watchredditdie, critical of censorship on Reddit]

In both (1) and (2), the speakers indicate similarly positive stances towards their interlocutor, but vary in their choice of stance marker to demarcate this stance (*insanely* vs. *incredibly*).

Importantly, sociolinguistic work has also argued that the *contexts* in which such markers appear is a critical aspect of stancetaking behavior (Kiesling et al., 2018; Bohmann and Ahlers, 2022). For example, sentences (2) and (3) use the same stance marker (*incredibly*), but in (3) the speaker evaluates their interlocutors in an especially negative light.

Sociolinguistic work posits that repeated stancetaking in a consistent way leads to stable associations of stance and identity (Eckert, 2008; Rauniomaa, 2003, cited in Bucholtz and Hall, 2005). For example, positive evaluations like (2) reinforce the identity of r/progresspics as a supportive community, while negative evaluations like (3) contribute to the critical tenor of r/watchredditdie. This illustrates that, in addition to stance marker preferences, stance contexts are crucial for connecting

stancetaking and community identity. However, while large-scale research has shown that variation in stance marker usage captures patterns of user cross-posting (Pavalanathan et al., 2017), to our knowledge, no large-scale work has considered how variation in stance contexts relate to community identity and cross-community engagement.

Here we address that gap by proposing a method for representing stance-relevant properties of a sentence that go beyond stance marker usage. Specifically, our **stance context representations** capture higher-level linguistic properties theorized to relate to stancetaking, including affect, politeness, and formality (Jaffe, 2009; Kiesling, 2009; Pavalanathan et al., 2017; Kiesling et al., 2018).

We focus our attention on the stance contexts in which **intensifiers** appear. Intensifiers, like *insanely* and *incredibly* above, are a subset of stance markers that emphasize that which they modify. Sociolinguistic work has shown that intensifier usage varies across linguistic and social contexts (Ito and Tagliamonte, 2003; Bolinger, 1972), which makes them ideal for studying community variation in stancetaking.

We first show that our stance context representations reveal aspects of community identity in a manner that complements both general textual representations, as well as stance marker preference representations. We then demonstrate how similarities in stance context relate to broader inter- and intra-community engagement patterns, including user co-participation across Reddit communities, and social factors of communities, including size, activity, loyalty, and density. Our work thus broadens the research linking community linguistic practices to social and network properties by considering higher-level factors beyond lexical variation. Future work can leverage these representations to shed light on additional aspects of platform dynamics, including community creation (e.g. are new communities sometimes created so that users may discuss the same topic but taking an alternative stance?) and user diversity (e.g. on Reddit, a platform dominated by men, do the stances communities take affect the participation and engagement of women and people with a (stereotypically) feminine linguistic style?).[1]

---

## 2 Related Work

To study the relationship between language and community identity online, work in computational sociolinguistics has begun to shift from relating language choices and macrosocial categories like age, gender, and race (Burger et al., 2011; Nguyen et al., 2013; Blodgett et al., 2016), to a more nuanced approach to identity, adopting the framework of communities of practice (Eckert and McConnell-Ginet, 1992; Holmes and Meyerhoff, 1999). Communities of practice put the focus on local identity categories that better reflect individuals' various group membership choices – e.g., presenting as a gamer, a sports fan, or an animal lover.

Much sociolinguistic work has explored the connection between linguistic variation and social network structure in communities of practice (e.g., Cheshire, 1982; Milroy, 1987; Eckert, 2000; Sharma and Dodsworth, 2020). For example, Eckert (2000) noted the interrelation between how high school "jocks" and "burnouts" varied in their linguistic patterns and in the communities they interacted with. Work of this kind has focused on variation within a single community or a small number of communities.

Online social media platforms support research relating linguistic and non-linguistic aspects of community identity at scale through analysis of *multi-community* settings (e.g., Hamilton et al., 2016; Israeli et al., 2022; Noble and Bernardy, 2022). For example, Noble and Bernardy (2022) show positive correlations between similarities in community linguistic practices and user-community co-occurrence patterns. Further work has shown that communities with more distinct words better retain users (Zhang et al., 2017), and tend to be smaller, denser, more active, and have more local engagement (Lucy and Bamman, 2021). Similar findings have been shown for communities with distinct word senses (Del Tredici and Fernández, 2017; Lucy and Bamman, 2021). We extend computational sociolinguistic work on community identity and engagement patterns by motivating and investigating variation of a different kind: variation in higher level properties of stancetaking.

Stancetaking is key to understanding how language variation reflects identity, because so many of a speaker's linguistic choices pertain to expression of stance (e.g., Jaffe, 2009). Moreover, much work has argued that, rather than linguistic forms being directly associated with social iden-

tities, linguistic forms are first associated with stances, which in turn are associated with social identities (e.g., Ochs, 1993; Du Bois, 2002; Rauniomaa, 2003; Bucholtz and Hall, 2005; Eckert, 2008). Thus, variation in higher-level properties of stancetaking needs to be given more attention in analyses of language and community identity.

Yet there has been limited computational work on stancetaking. Pavalanathan et al. (2017) studied variation in use of stance markers across online communities, and their word-based approach allowed them to study stance at scale. Kiesling et al. (2018) developed a more nuanced approach for representing stance, drawing on the framework from Du Bois (2007); however, the approach depended heavily on manual annotation, limiting analysis to a relatively small number of online communities. In our work, by automatically assessing stance-relevant properties of contexts, we are able to maintain the scale of analysis from Pavalanathan et al. (2017), while getting closer to the richness of representation from Kiesling et al. (2018).

## 3 Representing Stance Contexts

We study stancetaking – and stance contexts in particular – on Reddit, due to its nature as a multi-community platform with free-form commenting and rich engagement patterns among its communities (subreddits). We take intensifiers – a kind of stance marker – and the stance contexts in which they appear as our testbed. Intensifiers, like *insanely* and *incredibly* in (1)–(3) above, are adverbs that give emphasis to what they modify.

Intensifiers are an ideal domain for studying community variation. Extensive empirical work in sociolinguistics has established that intensifiers vary based on social factors (e.g., Ito and Tagliamonte, 2003; Tagliamonte and Roberts, 2005); moreover, they also exhibit much versatility in their contexts and undergo frequent change (e.g., Bolinger, 1972; Peters, 1994). While past computational work has studied semantic change in intensifiers over time (Luo et al., 2019; Samir et al., 2021), to our knowledge computational work has not examined community variation in intensifiers, and their context of use, as we do here.

Furthermore, intensifiers are a practical choice: Stance markers cover a range of grammatical classes (verbs, adjectives, interjections, and others), and focusing on intensifiers (which are adverbs), ensures that the variation we find in stance contexts

is not due to variation in the grammatical contexts in which the markers appear.

Here we describe how we identify a set of intensifiers and extract their contexts from a large set of communities (subreddits). We then explain how we represent these contexts with linguistic features relevant to stancetaking, presenting the features, how we calculate them, and how we use them to form a vector-based representation of each community in our dataset.

### 3.1 Extracting the Data

We start with the list of single-word intensifiers from Bennett and Goodman (2018, Studies 1a and 1b), as they are shown to express a range of degree of intensification. To ensure that these are appropriate to our study of stancetaking, we intersected this list of 89 intensifiers with a list of 1118 stance markers (including intensifiers and other words) found on Reddit. The latter list was created using the methodology of Pavalanathan et al. (2017), applying Wang2Vec (Ling et al., 2015) to expand a seed set of 448 stance markers released by Biber and Finegan (1989).[2] The Wang2Vec model is trained on Reddit data and used to extract words used in similar distributional contexts to the seed stance markers. After intersecting the two lists, we removed two words that occurred in a stopword list (*most* and *very*) to ensure we included informative words.

This procedure yielded a set of 38 intensifiers that serve as our stance markers. This stance marker set, and full details of our method for obtaining it, are described in Appendix A. In Appendix B, we show additional analyses conducted on a larger set of 252 intensifiers, and find that our methodology is robust to the set of intensifiers used.

To create our stance context representations of Reddit communities, we needed to extract the contexts of use of these intensifiers. We drew on Reddit data from all of 2019 and retrieved all comments from the top 10K subreddits by activity (as determined by Waller and Anderson, 2019) using the Pushshift data dumps (Baumgartner et al., 2020)[3]. We applied preprocessing as described in Appendix C, yielding 1.2B comments across the 10K subreddits.

From each comment, we extract any sentence that contains an intensifier from our list, so that

---

[2]Wang2Vec is an adaptation of Word2Vec that accounts for syntactic information.

[3]https://files.pushshift.io/reddit/comments/

our stance context representations will reflect the local semantic context of this stance marker. We retain only those sentences with exactly one usage of an intensifier, and whose length is at least 6 tokens (to contain sufficient content to reliably extract values for our linguistic features). To ensure sufficient and comparable data per community, we only perform analyses on communities with at least 10K such sentences, and use exactly 10K sentences from each (subsampling when necessary), resulting in 17.98M sentences across 1798 communities.

## 3.2 Stance Context Representations

Our aim is to represent the prototypical stance context of each community as a vector of feature values averaged across its 10K sentences. We first describe how we assess properties of sentences, and then explain how we combine these to generate community-level representations.

### 3.2.1 Properties of Stance Contexts

To capture salient properties of stance contexts, we use linguistic features known to relate to stance demarcation, including affect, politeness, and formality (Jaffe, 2009; Pavalanathan et al., 2017; Kiesling et al., 2018). Affect was broken down into the 3 features of valence (positivity), arousal (intensity), and dominance (level of control), following the VAD framework commonly used in psycholinguistics for representing emotions (Russell, 2003). These 5 features are particularly relevant here, as intensifiers are often used for affective impact, and vary widely in politeness and formality. Moreover, such features have robust methods for assessment at the sentence level.

For each feature, we create a model for inferring an appropriate value given a sentence represented using SBERT (Reimers and Gurevych, 2019). SBERT generates high-quality sentence embeddings shown to be useful for inferring sentiment, a high-level property of text similar to our stance context properties (Reimers and Gurevych, 2019). We evaluated the SBERT models for each feature on held-out portions of the human-annotated datasets and achieved accuracies comparable to prior work. We summarize model details below; full details can be found in Appendix D.

For VAD, we follow Aggarwal et al. (2020) and train a Beta regression model to predict each of these scores (constrained to the range $[0, 1]$) given the SBERT representations of the 20K words in the NRC-VAD lexicon (Mohammad, 2018). We

can then apply this model to assess the VAD of full sentences, based on their SBERT representations.

For politeness, we build a logistic regression model, adapting the method of Danescu-Niculescu-Mizil et al. (2013) to give normally-distributed continuous politeness scores by taking the log-odds of the predicted politeness probability, and then min-max scaling the values to the range $[0, 1]$.

For formality, we used the annotated corpus compiled by Pavlick and Tetreault (2016), focusing on the Answers and Blogs domains. The former generalizes well to stylistically-different domains (Pavlick and Tetreault, 2016), while the latter is the most similar domain to Reddit (Pavalanathan et al., 2017). We rescaled the formality annotations to the range $[0, 1]$ using min-max scaling, and then trained and selected a formality prediction model using 10-fold cross-validation.

### 3.2.2 Community-Level Representations

We apply each of the above models to the sentences in our dataset. We are interested in the properties of the *context* in which an intensifier is used, rather than properties of the intensifier itself. Thus we replaced the intensifier in a sentence with the $[MASK]$ token, and generated SBERT representations for these masked sentences. We then applied our models described above to find the valence, arousal, dominance, politeness, and formality values for each sentence. Table 1 shows example sentences that vary along the five properties.

Our aim was to average the values of the features over all sentences in a subreddit to yield a community-level representation – its prototypical stance context vector. However, we observed that the stance contexts within a subreddit may fall at both extremes of a scale (e.g., both many highly positive sentences and many highly negative sentences), and averaging would obscure such a pattern. This type of behavior seems natural, because intensifiers are often derived from extreme adjectives – those that express a property towards the extremes of a scale (Samir et al., 2021) – and such extremeness may itself be part of the stance tenor of a community.

We thus added an "extremeness" version of each feature introduced in Section 3.2.1 to our stance context representation, calculated by centering each feature at $0$ and computing its absolute value. Including the extremeness features resulted in each sentence being represented by a 10-dimensional vector. We then represent each com-

| Sentence | V | A | D | P | F |
|---|---|---|---|---|---|
| OMG, I'm so *[MASK]* sorry for your loss. | 0.20 | 0.52 | 0.33 | 0.80 | 0.17 |
| The books are *[MASK]* easy to read and damn good. | 0.85 | 0.35 | 0.70 | 0.58 | 0.39 |
| This is *[MASK]* insightful advice. | 0.77 | 0.43 | 0.66 | 0.64 | 0.79 |
| Not gonna lie, that's *[MASK]* lame | 0.24 | 0.44 | 0.31 | 0.28 | 0.10 |

Table 1: Example stance context representations for the marker *incredibly*, which has been replaced with the *[MASK]* token. The columns represent valence (V), arousal (A), dominance (D), politeness (P), and formality (F).

munity's overall stance context as the mean of these 10-D vectors of its 10K sentences.[4]

## 4   Additional Representations/Methods

Here we present two other community representations to which we will compare our stance context representation: one that captures community preferences for the stance markers, and the other simple overall textual similarity. We also show how we calculate community similarity using the three representations. Finally, some analyses look at groups of subreddits by topic, which are described here.

**Marker Preference Representations.**   To capture community preferences for particular stance markers, we used the methodology of Pavalanathan et al. (2017), applied to our 17.98M instances of intensifiers, for comparability to our stance context representations. We create a subreddit-marker count matrix and apply a positive pointwise mutual information (PPMI) transformation to the matrix. We then apply truncated SVD and retain the top 11 dimensions (based on elbowing in a Scree plot).

**Textual Representations.**   We create textual representations to show that our stance context representations capture more than just textual similarity between communities. Following Lucy and Mendelsohn (2019), we construct a tf-idf weighted subreddit-word matrix from our 17.98M comments, with a vocabulary size of 158K. We then apply truncated SVD and retain the 7 most informative dimensions (based on elbowing in a Scree plot).

**Computing Community Similarity.**   We use cosine similarity to compute how similar two communities are on a particular representation. Let $R$ be one of our three representations, $c, c'$ be two communities (subreddits), and $R_c$ be the embedding for

community $c$ in representation $R$. Then:

$$Sim(R, c, c') = \text{cos\_sim}(R_c, R_{c'})$$

**Community Topic Groups.**   To investigate general trends in how communities vary with respect to each of our three representations, we group our subreddits by topic, using the r/ListofSubreddits[5] categorization, as in Lucy and Bamman (2021). These groups yield an interpretable community structure that broadly reflects commonalities in user interests.[6] The list divides Reddit into 12 major topics, with further subdivisions.

We designate any subtopics with at least 50 subreddits in our dataset as their own topic. For any unassigned subreddits that start with "ask", or include cue words such as "advice" or "questions", we add them to the Discussion subtopic. This results in 1228 subreddits assigned to 15 topics, and 570 subreddits without a topic label. The topics and number of subreddits assigned to them are shown in Appendix E.

**Computing Topic Group Similarity.**   To assess similarity of two topic groups, we find the mean pairwise similarities for all subreddits in each of the topics. Let $T_A, T_B$ be the sets of communities for topics $A$ and $B$. We compute the topic similarity $TS(R, T_A, T_B)$ for a representation $R$ as:

$$TS(R, T_A, T_B) = \frac{\sum_{c \in T_A} \sum_{c' \in T_B} Sim(R, c, c')}{|T_A||T_B|}$$

## 5   Comparing the Three Representations

We show how the 3 kinds of community representations (textual, stance context, and marker preference) capture distinct facets of community identity.

---

[4]To avoid anisotropy in our vector space, which resulted in almost all communities having near perfect cosine similarity with each other, we applied Z-score normalization to each of the 10 features.

[5]https://www.reddit.com/r/ListOfSubreddits/wiki/listofsubreddits/, updated as of April 2023.

[6]The topics are specified by Redditors, and capture subreddit similarity based on perceptions of what may drive people to engage with communities, rather than simple textual overlap. For instance, both r/law and r/learnart are under Education, but r/learnart is most textually similar to subreddits like r/crossstitch and r/artistlounge, which are under Hobbies.

| | |
|---|---|
| Textual vs. Stance Context | 0.52*** |
| Textual vs. Marker Usage | 0.39*** |
| Stance Context vs. Marker Usage | 0.37*** |

Table 2: Pearson correlations of our subreddit-level representations. *** indicates significance at p < 0.001.

We find that they do not correlate highly with each other; moreover, they can reveal subtle differences between pairs of seemingly similar subreddits.

## 5.1 Pairwise Correlations

For each representation, we compute the community similarity of each of the 1.6M pairs of subreddits, and find the pairwise Pearson correlations between these values. Table 2 shows that the representations have only low to moderate positive correlations. This provides preliminary evidence that we have representations that capture three distinct facets of subreddits: the content they discuss, the stances they tend to take with respect to this content, and the stance markers they use to demarcate these stances. In the next section, we illustrate how these representations capture distinct patterns across the multi-community landscape.

## 5.2 Topic-Topic Similarities

To visualize the differences in what these three representations capture, we compute $TS(R, T_A, T_B)$ for all pairs of topics $A$ and $B$ (excluding the Other and General categories as they lack a topical focus).[7] The plots in Figure 1 show much variation across the three representations. To dig deeper into the nature of this variation, our qualitative analysis focuses on three pairs of topics that illustrate how communities can vary in their degree of textual, stance context, and marker preference similarity; see Figure 2. Further visualizations comparing the differences between representations can be found in Appendix F.

The Animals and Humor topics have a not uncommon pattern in which they are moderately to very similar across all three representations. The linking of animals and humor is not surprising, although the range of expressed stances is fairly broad. For example, r/animalsbeingderps, r/animalsbeingjerks, and r/animalsbeingbros involve observing animal behavior that is either awkward, mean, or kind, respectively; their high sim-

ilarity with humor subreddits indicate that these may be varied aspects of humor as well.

The other two pairs of topics, Politics/Education, and Sports/Video Games, have low to moderate textual similarity, but show very high similarity in either stance context or marker preference. The Politics and Education topics have somewhat higher formal stance contexts relative to other subreddits, which may arise from the local identity categories that surface in these kinds of communities, as individuals may seek to position themselves as intelligent. This higher formality leads to very consistent stance marker use, as 8 of their top 10 most preferred markers overlap, including *largely*, *enormously*, *wholly*, and *exceedingly*. These markers are 4 out of the top 5 markers used to demarcate formal stances across our whole dataset. This example highlights how community stance similarity on a particular dimension can guide marker preference similarity.

One of the starkest contrasts across the three representations is between the Sports and Video Games topics. Despite having dissimilar textual representations, this pair of topics is among the most similar with respect to stance contexts (in fact, there is less variation between the stances of these communities than within most topics on the diagonal). Analysis reveals that they are similar on most dimensions, but are noticeably distinct from the rest of the platform with respect to their high arousal and less extreme politeness. The high arousal (intensity) of these topics makes sense given the competitive, game-oriented nature of the subreddits in both these groups. However, despite this similarity in stance contexts, the Sports and Video Games subreddits have low marker preference similarity, highlighting that communities that take similar stances may differ greatly in how they choose to demarcate these stances.

## 5.3 Discussion

These findings reveal the differences in textual, stance context, and marker preference representations. Each provides a distinct glimpse into facets of community identity, and using all three simultaneously provides further insights into axes on which Reddit communities tend to vary. Examining the Politics and Education topics shows how similarity on just a single dimension of stance can be informative as to marker preference similarity, as communities seek to demarcate stance in a way that

---

[7]Since the range and variance of similarity values varies across representations, we apply Z-score normalization to the set of topic-topic similarities.

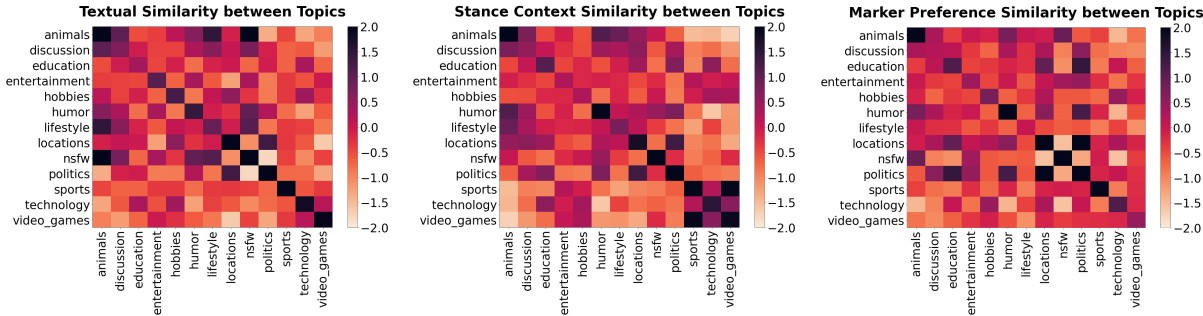

Figure 1: Patterns of topic–topic similarity for our three community representations. Dark cells indicate topics that are more similar to each other than average, while light cells indicate topics that are more dissimilar to each other than average. Cells on the diagonal represent the variation within each topic.

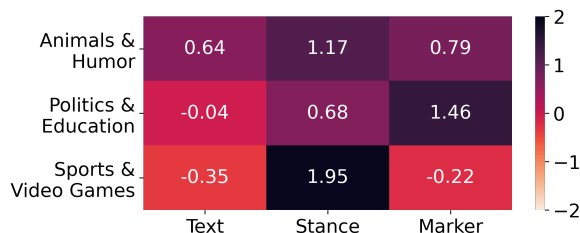

Figure 2: Topic-topic similarities for each of our three representations for pairs of interest. Colors and z-scores are the same as in Figure 1.

is consistent with particular traits. Our final example illustrates that communities with almost no similarity in their content (Sports and Video Games) can be extremely similar in how their stances relate to that content (highlighting a critical difference between stance contexts and textual similarity), but may yet demarcate their stances differently (highlighting the difference between stance contexts and stance markers). Overall, our results show that our three representations are distinct yet complementary, and provide a holistic view of community variation in language and community identity.

## 6 Stance and Engagement Behavior

Our analysis in the preceding section has established that stance context is distinct from both textual similarity and stance marker preference, and has shown preliminary evidence that these differences allow us to capture high-level properties of communities that relate to social identity. In this section, we demonstrate how the two aspects of stancetaking, stance context and stance marker preferences, relate to community engagement patterns on Reddit. We first examine how well they predict subreddit cross-posting patterns. We then explore how community distinctiveness in both average

stance context and stance marker preferences relates to community structure. These findings illustrate the explanatory power offered by stance-based representations, particularly stance context representations, when investigating inter- and intra-community dynamics online.

### 6.1 Cross-Posting

Core to platforms like Reddit is that people may engage with a number of communities of practice. Among the many reasons users may cross-participate in such communities, we hypothesize that users are attracted to communities that have similar linguistic practices – in particular, those that are similar in how their members express stance. Pavalanathan et al. (2017) showed evidence of stance demarcation patterns predicting user cross-posting patterns, and we extend this to looking at community similarity in stance marker contexts.

Our classification task follows the same procedure described in Pavalanathan et al. (2017), adapted for the top 100 most popular subreddits in our dataset by commenting volume. This results in 429 low cross-posting pairs and 353 high cross-posting pairs. For each representation and a given subreddit pair $(c, c')$, we use a logistic regression model to predict whether $c$ and $c'$ have high or low cross-posting using $Sim(R, c, c')$. (We also replicated the model used in Pavalanathan et al. (2017) that treats each vector dimension as a separate predictor and found a similar pattern of results to our cosine similarity metric, but with lower accuracies.)

To evaluate our models, we compare their accuracy at predicting cross-posting using the average accuracy of a 10x10 repeated cross-validation,[8]

---

[8]Similarly to Boleda et al. (2012), we use the corrected resampled t-test developed by Nadeau and Bengio (2003) for performing statistical tests, as the lack of independence

| Representation | Accuracy |
|---|---|
| Marker preference similarity | 75.1% |
| Stance context similarity | 82.3% |

Table 3: Accuracy in predicting user cross-posting.

|  | Size | Activity | Loyalty | Density |
|---|---|---|---|---|
| Marker Pref. | 0.00 | -0.02 | **0.20** | 0.01 |
| Stance Context | **0.21** | **-0.36** | **-0.34** | **-0.34** |

Table 4: Spearman correlations of social factors and distinctiveness; bold indicates significance at the $\alpha = 0.05$ level using Bonferroni correction.

shown in Table 3. Using only our 38 intensifiers, the marker preference representation performs surprisingly well, substantiating the importance of intensifiers in communicating stance. However, our stance context representations perform even better at predicting cross-posting (t = 4.28, p < 0.001, Cohen's $d = 1.79$), confirming the key role of stance context features in community linguistic practices.

To determine which features of stance affect how users cross-participate on subreddits, we also examined the standardized coefficients of a regression model that treats each predictor separately. We found that 7 of our 10 stance context features were significant predictors of cross-posting, including 4 of the 5 extremeness features. Similarity in formality and extremeness of valence were especially predictive of cross-posting. The full results can be found in Appendix G.

Thus we show here that variation in stance contexts is indeed predictive of the community membership decisions that Redditors make. Despite only capturing how communities use intensifiers, both marker preference similarity and stance context similarity do reasonably well at predicting cross-posting patterns on Reddit. Our stance context representations obtained 7% higher accuracy on the task, suggesting that higher-level contextual properties related to stance are particularly important for shaping user participation patterns. Furthermore, our interpretable stance context properties allowed us to study the aspects of stancetaking that most shape co-participation online. Each of these findings helps shed light on how people engage across multiple communities of practice.

### 6.2 Distinctiveness in Stancetaking Style

Our cross-posting analysis confirmed that the ways people express stance are related to how they choose to engage across multiple communities. We now turn to the extensive sociolinguistic findings referenced in Section 2 that linguistic practices also interrelate with community structure. For example, Lucy and Bamman (2021) showed that the use

of specialized vocabulary varies with four social network properties in online communities. Specifically, Lucy and Bamman (2021) found that communities with more distinctive use of words and of word senses are smaller (fewer users), more active (higher average participation among users), more loyal (more users whose participation is focused in that community), and more dense (more interactions among users). We sought to explore whether stancetaking patterns similarly in online communities – do smaller, more active, more loyal, and denser communities develop a more distinctive manner of expressing stance?

For this question, we explore distinctiveness in stancetaking style with respect to both stance contexts and marker preferences, and relate these to the four social factors explored in Lucy and Bamman (2021).[9] We conceptualize distinctiveness $D(c, R)$ of a community $c$ as how **dissimilar** it is from all other communities on average, using representation $R$. Let $C$ be our set of 1798 communities:

$$D(c, R) = \frac{\sum\limits_{c' \in C} (1 - Sim(R, c, c'))}{|C|}$$

Then, for each representation $R$ (stance context and marker preference), we perform correlations between the $D(c, R)$ values for all 1798 communities, and the values of each of the four social factors calculated for the communities; see Table 4.

First, we see that distinctiveness in marker preference is significantly correlated with loyalty; this is consistent with the finding of Lucy and Bamman (2021) that more loyal communities have more distinctive vocabulary usage. However, in contrast to Lucy and Bamman (2021), distinctive usage of stance markers is not correlated with any other network properties. Presumably the set of stance markers we consider are general enough vocabulary items that their preference patterns are not

---

between training sets across folds results in an inflated Type I error for the standard paired t-test (Dietterich, 1998).

[9]We adopted the formulas used by Lucy and Bamman (2021) for these factors, described in Appendix H.

highly associated with how users engage within a community.

Second, and more notably, stance context distinctiveness patterns in a manner *inverse* to what Lucy and Bamman (2021) found for distinctiveness of vocabulary.[10] Communities that are more distinct in their stance contexts tend to be larger, less active, less loyal, and less dense. To investigate what drives stance context distinctiveness to pattern in this way, we examine the communities by topic grouping. Specifically, we find that of the 303 least distinctive quartile of communities that have a topic assigned, 41% are Sports communities and 26% are Video Games communities. This is coherent with our results in Section 5, where we showed that despite having low textual similarity, Sports and Video Games communities have rather high stance context similarity. Since they form a sizable collection of subreddits, their mutual similarity leads to them having among the lowest average distinctiveness of our communities.

Moreover, among the 13 contentful topic groups (excluding General, Other, and Not Assigned), we find that Sports communities tend to be the smallest, densest, most loyal, and most active overall, while Video Game subreddits are larger but also among the highest in density, loyalty, and activity. Qualitatively, we suspect Sports and Video Game communities foster this degree of local engagement given allegiances people have to their preferred teams and games.

Thus, our findings on distinctiveness in stancetaking context reveal novel aspects of the relationship between linguistic practice and community structure. Because Sports and Video Game communities are highly represented on Reddit overall, it's not surprising that these break down into a large number of subcommunities that are more topically fine-grained (on particular sports, teams, and games). These communities will then be among the smaller, denser, more loyal, and more active subreddits, but because they have stance context properties in common, these social network properties are associated with the least distinctive communities in stance context.

Furthermore, the contrast with distinctiveness in vocabulary – both our marker preferences here and specialized vocabulary in Lucy and Bamman (2021) – allow us to view Reddit platform dynam-

---

ics and community structure from a different lens than previous work. Our findings illustrate that, as higher level properties of linguistic practice, stancetaking contexts may hold in common across different detailed communities. Our results overall highlight the importance of investigating the rich and varied linguistic means for expressing community identity.

## 7   Conclusion

We extend work on community variation in stancetaking by constructing stance context representations using theoretically-motivated linguistic properties of stance. Since stancetaking plays a key role in linguistic variation (Jaffe, 2009), considering these higher level properties of stance may help shed light on why community members present themselves in particular ways. We showed that our stance context representations capture aspects of community linguistic identity distinct from existing methods. Furthermore, we found that these representations were related to inter- and intra-community engagement patterns.

Our stance contexts approach also allowed us to reveal "mega" communities: those like Sports and Video Games that share stancetaking properties while having numerous fine-grained subcommunities within them. Further work with stance contexts may help shed additional light on the nuances of mega community structure: in addition to establishing communities for different subtopics (such as different video games), are "spin-off" communities (cf. Hessel et al., 2016) sometimes created so that users may discuss the same topic while taking an alternative stance (e.g. r/pokemontrades vs r/casualpokemontrades)?

Additionally, by showing that higher level properties of stance relate to community identity and engagement patterns, we pave the way for much richer models of the reasons that people are attracted to, or discouraged from, online participation in various groups. For example, on Reddit, a platform dominated by men, high-arousal stances are the most common due to the overrepresentation of Sports and Video Games communities. Future work could explore whether other kinds of stances (e.g. more polite or higher valence stances) lead to increased participation and engagement from women and people with a (stereotypically) feminine linguistic style.

---

[10] We replicated our results on the communities from Lucy and Bamman (2021) with similar results; see Appendix I.

## 8 Limitations

We first discuss limitations related to our specific methodological choices, and then discuss limitations of our approach more generally.

### 8.1 Methodological choices

First, we studied the contexts where speakers use intensifiers – a kind of stance marker – but we considered only 38 intensifiers in our main analyses. As a follow-up, we ran additional experiments that included a set of 252 intensifiers from Luo et al. (2019) across 448 of our communities selected for another project. These experiments confirm that both our cross-posting and social factors results hold for a much larger set of intensifiers. Full details of the results can be found in Appendix B.

However, it may be fruitful to replicate our analyses with a larger set of stance markers (including markers beyond intensifiers), such as the set of 812 stance markers considered by Pavalanathan et al. (2017). This would allow us to ensure our insights hold for stance markers generally.

There are also limitations to our method for assessing stance-relevant properties. We focused on valence, arousal, dominance, politeness, and formality, but other linguistic properties have been identified as relevant to stance contexts, including subjectivity and certainty (Kärkkäinen, 2006; Kiesling et al., 2018). Future work on stancetaking may benefit from the development of methods for assessing these properties.

### 8.2 General approach

There are also some limitations of our stance contexts approach more generally. First, our approach to assessing stance properties doesn't account for community-level semantic variation. For instance, work has shown that the same word can vary in sentiment, depending on the community (Hamilton et al., 2016; Lucy and Mendelsohn, 2019) . Because our approach for assessing stance-relevant properties depends on context words (which can vary in meaning across communities), it may miss out on some community-specific meanings of stance contexts.

Relatedly, our approach doesn't account for variation in how stances are communicated over time; the datasets we use to train our models for assessing stance-relevant properties (Mohammad, 2018; Danescu-Niculescu-Mizil et al., 2013; Pavlick and Tetreault, 2016) are 5–10 years old, and likely miss

out on novel words or word senses used in online contexts, and their effect on our five linguistic property values. Future research should work to understand how the stance-relevant properties of contexts vary across communities and over time.

Furthermore, although we drew on literature from sociolinguistics and linguistic anthropology in developing our stance contexts representations, our approach does not capture the full richness of stance as understood in these fields. For example, the highly influential stance triangle framework from Du Bois (2007) theorizes that stancetaking involves a speaker evaluating something (a "stance object"), and that – in so doing – the speaker positions themself relative to their interlocutor. Because these components of stance are sometimes mentioned in the sentential context (which we use to generate our stance representations), we may implicitly capture them in some cases. However, within our framework, it is not straightforward to compare these components of stance to human intepretations (cf. Kiesling et al., 2018).

Related to this, our method only captures local aspects of stance contexts within the sentence in which intensifiers were used. However, work on stance contexts has discussed how the sequential context of the preceding utterances in a dialogue shape the construction and interpretation of one's stance (Kiesling et al., 2018; Bohmann and Ahlers, 2022). For example, the preceding utterances could inform whether a stance is interpreted as sarcastic or not. Future research should work to develop even richer representations of stance, which are also scalable.

### 8.3 Data Access

One additional limitation is related to the future accessibility of our data, as the Pushshift Data Dumps we used to extract Reddit data are no longer available at their original link (as of May 2023). Future research will need to use the official Reddit API to access the data we used.

## 9 Ethical Considerations

User privacy is a concern inherent to using online data. The data we use was public when collected by Baumgartner et al. (2020), and we take care to remove data from any individuals that had deleted their accounts at any point prior to the data's collection, adhering as best we can to a user's right to be forgotten.

Mohammad (2022) mentions that in cases where one automatically infers emotional properties (such as VAD scores), it is critical to be cognizant that users may not want their data analyzed, as well of the potential harms of associating such information with individual users. Furthermore, the automatic assessment of emotional properties of text may not match the intended affect of the user, and this problem may be more pronounced for users who are part of marginalized groups (Mohammad, 2022). With these concerns in mind, we remove user-level information prior to extracting values for our linguistic features, such that our downstream analyses have no connection between individual users and the emotional aspects of their language. Our textual and marker preference representations also make no use of user identifiers. We only make use of user-identifying information for computing community engagement behavior, and only release information about the aggregated statistics per community. Although the sentences we use in our dataset were created by individual users, we mitigate user privacy concerns as best as possible in the ways listed above.

Another ethical concern pertains to the amount of compute power we used for our analyses, particularly for data extraction and creating our sentence embeddings (see Appendix J). For the latter process, we minimized unnecessary computation by computing the embeddings for each sentence only once. This means that we did not need to recompute embeddings for each of our five linguistic properties, and that we can extend our methodology to include other linguistic properties without needing to use any more GPU power for creating embeddings. To mitigate additional compute costs for researchers seeking to replicate our work, we make our sentence-level stance representations public.

## Acknowledgements

We acknowledge the support of NSERC of Canada (through grant RGPIN-2017-06506 to SS), as well as the support of the Data Sciences Institute, University of Toronto (through a Catalyst Grant to SS and JW).

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

## A Constructing the Stance Marker and Intensifier Lists

To extract a set of stance markers common to Reddit, we replicated the lexical expansion procedure used by Pavalanathan et al. (2017). We initially focused our analyses on Reddit data from 2014, and performed lexicon expansion using this dataset.

| | |
|---|---|
| absolutely | plain |
| amazingly | pretty |
| awfully | quite |
| considerably | really |
| decidedly | remarkably |
| downright | seriously |
| enormously | significantly |
| exceedingly | simply |
| exceptionally | surprisingly |
| excessively | suspiciously |
| extraordinarily | terribly |
| extremely | totally |
| fantastically | tremendously |
| hugely | truly |
| incredibly | unbelievably |
| insanely | unusually |
| intensely | utterly |
| largely | wholly |
| noticeably | wonderfully |

Table 5: List of intensifiers used in this study.

We started with the seed set of stance markers released by Biber and Finegan (1989), which includes 448 markers in total. For a separate project, we required the stance categories that this corpus assigns markers to (e.g. affective stance markers, emphatic stance markers, etc.), so this set was more appropriate than the stance markers released in the Switchboard corpus (Jurafsky et al., 1998), which does not contain category-level information.

From this initial set of seed stance markers, we search for other stance markers on Reddit used in similar distributional contexts. To do so, we use Ling et al. (2015) (an adaptation of Word2Vec that accounts for syntactic information), following the method of Pavalanathan et al. (2017). Unlike Pavalanathan et al. (2017), we train Wang2vec on all words that occur at least 5 times in our sample of 25M comments from 2014, resulting in a vocabulary of size 1.2M. We adjust their thresholds for similarity, extracting all candidate words having a cosine similarity of at least 0.85 with one of our seed markers, and a general frequency of at least $10^{-6}$.

This process resulted in 1118 stance markers in total, of which 40 are intensifiers found in Bennett and Goodman (2018). We remove the two stop words *most* and *very* to arrive at our final set of 38 intensifiers, shown in Table 5.

## B  Expanded Analyses

In our main analyses, we focused on a relatively small number of intensifiers (38), which may raise concerns about whether our findings generalize to other intensifiers. To address this, we performed the same cross-posting and social factors analyses on the 252 intensifiers from Luo et al. (2019) on a subset of 448 subreddits (selected for a different project).

Of our 38 stance markers, 28 are found in this set of 252 markers. The 10 markers that are not in the set of 252 markers are: *downright*, *fantastically*, *incredibly*, *largely*, *plain*, *pretty*, *quite*, *really*, *suspiciously*, and *unbelievably*. We repeat our analyses on the 252 markers collected for the new dataset, as well as on the 28 markers both datasets have in common.

Across both our original and additional analyses, our stance context representations achieve very consistent results. We expand on each analysis in the sections below.

### B.1  Cross-Posting Results

For the cross-posting analysis, we recompute high and low cross-posting pairs using the same methodology as in the main text, but for the set of 448 communities in this analysis. Table 6 shows the results, where the leftmost column corresponds to the results in the main text. The stance context representations significantly outperform the marker preference representations in all cases, at $\alpha = 0.05$ using the same 10x10 cross-validation procedure used in the main text. This suggests that stance context representations are consistently better at predicting cross-posting than the marker preference similarity representations.

### B.2  Social Factors Analysis

We repeat the social factors analysis for the stance context representations using the same procedure outlined in the main text. In the dataset with 448 communities, distictiveness is calculated only with respect to these 448 communities. Table 7 shows our results.

As in the case of the cross-posting analysis, the correlations between stance context distinctiveness and the four social factors also shows the same pattern as in the main text results across all the new analyses, albeit with some reduction in the power and magnitude of correlation when only using the 28 overlapping intensifiers. This suggests that our

representations, as well as our overall findings, are robust to the selection of intensifiers.

## C  Data Preprocessing

We apply preprocessing to the comment text extracted from Reddit, including: We replace all links with a $[LINK]$ token, replace all mentions of another username with a $[USER]$ token, and replace irregular unicode characters, extraneous parentheses, and newlines with whitespace. Malformed quotation marks and apostrophes are also replaced with the appropriate token. We then strip tokens used to indicate bold or italicized text and replace multi-whitespace characters with a single character.

At the author-level, we remove any comments written by deleted authors, AutoModerators, and account names that end in "bot", regardless of case.

## D  Details of Models to Infer Linguistic Properties

To create SBERT representations of sentences in our data, we use `bert-large-nli-mean-tokens`, as it performs the best on the STS task without fine-tuning (Reimers and Gurevych, 2019).

For valence, arousal, and dominance, we have access to the NRC-VAD lexicon (Mohammad, 2018), which provides values of these features for 20K words, but we need a model that assigns such values to sentences. We follow the method of Aggarwal et al. (2020) by training Beta regression models on SBERT representations of words from the NRC-VAD lexicon, and then applying those models to our Reddit sentence data. We adjusted the methodology of Aggarwal et al. (2020) by training the models on 80% of the data, stratified over quintiles of each of the VAD scores. This stratification ensured that our training data was approximately uniform across the $[0, 1]$ interval. We repeated this procedure 10 times and chose the model that performed best for each feature. Our best models achieve Pearson correlations of $0.85$, $0.77$, and $0.80$ on their held-out sets for valence, arousal, and dominance respectively, comparable to those found by Aggarwal et al. (2020).

For politeness, we followed the methodology of Danescu-Niculescu-Mizil et al. (2013) to build a logistic regression model to predict whether a given sentence is polite. We adapted their approach to give normally-distributed continuous politeness scores by taking the log-odds of the predicted politeness probability, and then min-max scaled the

| | | Marker Preference Similarity | Stance Context Similarity |
|---|---|---|---|
| **Original Dataset (n=1798)** | **38 markers** | 75.1% | 82.3% |
| | **28 markers** | 70.6% | 81.0% |
| **New Dataset (n=448)** | **252 markers** | 76.5% | 79.8% |
| | **28 markers** | 73.3% | 78.5% |

Table 6: Cross-posting results for both our original and new datasets with the full and intersected marker lists.

| | | Size | Activity | Loyalty | Density |
|---|---|---|---|---|---|
| **Original Dataset (n=1798)** | **38 markers** | **0.21** | **-0.36** | **-0.34** | **-0.34** |
| | **28 markers** | 0.18 | **-0.36** | **-0.26** | **-0.30** |
| **New Dataset (n=448)** | **252 markers** | 0.14 | **-0.32** | **-0.34** | **-0.33** |
| | **28 markers** | 0.13 | **-0.35** | **-0.29** | **-0.31** |

Table 7: Spearman correlations of social factors and stance context distinctiveness in our datasets; bold indicates significance at the $\alpha = 0.05$ level using Bonferroni correction applied to 8 tests.

resulting values to the range $[0, 1]$. We trained and tested our models on both their Wikipedia corpus and their StackExchange corpus, performing both in-domain and cross-domain tests. In-domain tests were conducted using 3x10 cross-validation, and the cross-domain tests were conducted by training the model on the entire training set and testing on the entire testing set. We found that the model trained on the Wikipedia text performed better than the model trained on the StackExchange text, with in-domain and cross-domain accuracies of 84.1% and 65.2% respectively, comparable to the models in Danescu-Niculescu-Mizil et al. (2013).

For the formality model, we selected the best model after 10-fold cross validation. The Spearman correlation of our best model with its hold-out set was 0.76, higher than those found in Pavlick and Tetreault (2016).

## E Subreddit Topic Groups

The list of topic groups we used, and the number of our 1798 subreddits assigned to each, are shown in Table 8.

## F Topic-Topic Differences in Representations

To further visualize differences in how topics differ across our three representations, we show the graphs in Figure 3. These graphs were generated by subtracting the values in each cell of Figure 1 between the corresponding heatmaps. For instance, the top-left cell of the leftmost subfigure of Figure 3 can be interpreted as follows: subreddits in

| Topic | # Subreddits |
|---|---|
| Entertainment | 176 |
| Video Games | 159 |
| Lifestyle | 147 |
| Sports | 142 |
| Discussion | 109 |
| Other | 106 |
| Locations | 96 |
| Hobbies | 94 |
| Education | 69 |
| Technology | 50 |
| Politics | 21 |
| Humor | 19 |
| General | 18 |
| NSFW | 15 |
| Animals | 7 |

Table 8: Number of subreddits per topic.

the Animals topic are more similar to each other according to textual representations than stance context representations.

## G Stance Properties and Crossposting

To assess each feature's individual contribution to predicting crossposting, we used a logistic regression model with 10 predictors: the differences between a pair of communities on a particular stance property dimension. To improve our model for this task, we unit-normalized each community stance context representation. The coefficients are shown in Table 9.

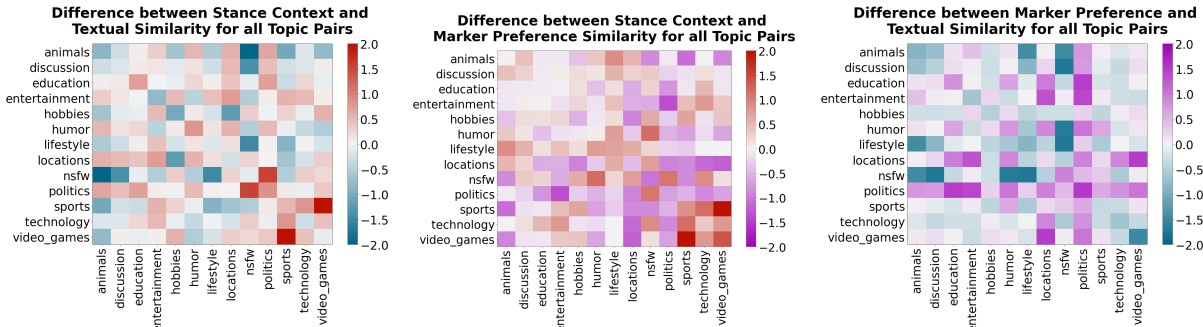

Figure 3: Patterns of topic–topic differences for each pair of representations. Darker cells represent a greater difference between the pairs of representations. Blue cells are those that are more similar with respect to text than other representations, red cells are more similar with respect to stance context than other representations, and purple cells are more similar with respect to marker preference than other representations.

|  | Raw | Extreme |
|---|---|---|
| Valence | -2.85** | -4.04*** |
| Arousal | -1.83 | -1.13 |
| Dominance | -0.53 | -2.32* |
| Politeness | -2.75** | -2.74** |
| Formality | -7.80*** | -3.04** |

Table 9: Feature importance at predicting cross-posting. Asterisks indicate significance at 0.05 (*), 0.01 (**), and 0.001 (***), respectively.

|  | Size | Activity | Loyalty | Density |
|---|---|---|---|---|
| Stance Context | **0.35** | **-0.42** | **-0.42** | **-0.46** |
| Marker Preference | -0.08 | 0.08 | **0.26** | 0.13 |
| Vocab. L&B,2021 | **-0.60** | **0.58** | **0.70** | **0.69** |

Table 10: Spearman correlations of social factors and distinctiveness for the 418 communities that overlap between our dataset and that of Lucy and Bamman (2021); bold indicates significance at the 0.05 level using Bonferroni correction.

## H  Computing Social Factors

As mentioned, we adopt the formulas used by Lucy and Bamman (2021) for computing the social factors of size, activity, density, and loyalty. All data is gathered from the set of 1.2B comments written in 2019.

For community size, we use the number of distinct users that have ever posted in a community. For user activity, we divide the number of total comments in a community in 2019 by the community size. To compute network density, we first construct an undirected parent-reply network for the top 20% of users by commenting volume in 2019. For each community, we then compute the density as the number of users who have an edge between them, dividing by the total number of possible edges in the graph. Finally, to compute loyalty, we find the proportion of users in a community that have at least 50% of their top-level comments in 2019 in a particular community.

## I  Social Factors Analysis for Vocabulary Distinctiveness

In past work, Lucy and Bamman (2021) examined how social factors relate to distinctiveness in vo-

cabulary. We wanted to build on their work by exploring the role of contextual properties beyond individual words and word senses, so in our main analyses, we focused on our novel contribution of how the social factors relate to distinctiveness in aspects of stancetaking.

Here, we directly compare how stance context distinctiveness and vocabulary distinctiveness relate to social factors. To do this, we use the data released by Lucy and Bamman (2021) and compute vocabulary distinctiveness scores for the 418 communities that our dataset has in common with theirs.[11]

The results for our stance context and marker preference distinctiveness, as well as vocabulary distinctiveness, are shown in Table 10. For this set of subreddits, we find the same pattern of results for both of our distinctiveness measures as on the full set of our subreddits, and we see the same pattern of results for vocabulary distinctiveness as reported in Lucy and Bamman (2021) for their full set of sub-

---
[11]Link to Github for Lucy and Bamman (2021): `https://github.com/lucy3/ingroup_lang`

reddits. Here we also found that Sports and Video Games subreddits are again (as in our full results) the most represented topics among communities in the lowest quartile of stance context distinctiveness (48% and 31%, respectively). Conversely, those topic groups are also most represented in the highest quartile of vocabulary distinctiveness (38% and 23%, respectively).

## J  Hardware

To extract our data, we used 12 CPUs in parallel, each of which took roughly 8 hours to extract data from the Pushshift data dumps. For computing sentence embeddings, we used two Nvidia Titan X GPUs, which processed about 1.1M sentences per hour (in batches of size 32). This took roughly 8 hours per GPU, for a total of about 16 GPU hours.