# OpenReview forum: "Investigating Online Community Engagement through Stancetaking"
_EMNLP/2023/Conference — EMNLP 2023 Findings_

### Official Review · Reviewer_5R1c · 2023-07-31

**Soundness:** 3

**Excitement:**

4: Strong: This paper deepens the understanding of some phenomenon or lowers the barriers to an existing research direction.

**Missing References:**

Inducing Domain-Specific Sentiment Lexicons from Unlabeled Corpora, Hamilton et al. EMNLP 2016.

**Paper Topic And Main Contributions:**

This paper investigates how stancetaking in communities can shed light on community identity and language at scale in nearly 1.8k communities on Reddit. The authors operationalize stancetaking by creating “stance context” vector representations of communities. First, they mask out intensifiers (e.g. “incredibly”) in sentences and generate the Sentence BERT (SBERT) representation of these sentences. Then, they reduce these SBERT representations into smaller vectors containing dimensions related to valence, arousal, dominance, formality, and politeness using various predictive models, and then aggregate these to the community-level by taking their average. The authors show that these representations are distinct from prior ways of measuring a community’s stance or language, and more accurate at predicting user cross-posting behavior. Finally, the authors show that stancetaking relates to user loyalty, activity, density, and size in directions that differ from past work on linguistic distinctiveness.

**Questions For The Authors:**

Since the paper frames itself as focusing on stancetaking, one may expect it to investigate a variety of stance markers. Would this study be extendable to other types of stance markers beyond intensifiers? Why did the authors restrict it to just intensifiers?

Why is the text-based representation of communities left out of the user cross-posting prediction analysis (Table 2)?


**Reasons To Accept:**

This paper is generally well written. For example, the authors’ use of a few key examples in the introduction was a nice way to provide background on the nuances of operationalizing stancetaking. I appreciate that the authors grounded the motivation behind their approach using research from sociolinguistics, and took care to outline how their work fills gaps in prior computational work (e.g. lines 174-178).

Overall, this paper could benefit others’ ongoing research on how language relates to community identity at scale.


**Reasons To Reject:**

Lines 210-224, which describe the process for deciding what stance intensifiers to use in this study, was confusing as to why the authors made the decisions that they did. The authors start with a wide pool of potential markers (89 intensifiers intersected with a list of 1000 stance markers) but then at the end they keep only 38. Why were so many intensifiers removed? The main text should be able to stand alone from the Appendix, but I also looked in the Appendix and it raised even more questions. For example, why use Bennett and Goodman (2018) to filter out which intensifiers to include?

In the paper, the authors reduce Sentence BERT vectors into 10 features related to affect, formality, and politeness predicted from these vectors. I’m not fully convinced as to why politeness, affect (measured by NRC-VAD), and formality are key to measuring stancetaking. Why not just take the average of SBERT vectors instead of reducing those vectors down to these narrow linguistic properties? It’s possible that the 10-dimensional feature vectors offer interpretability, but it’s also possible that SBERT vectors may include richer information that would be helpful for understanding what concepts communities tend to take stances on.

When it comes to weighing the potential contribution or impact of this paper, I would emphasize Section 6 (relating stance to engagement behavior) more so than Section 5 (where the authors argue that intensifiers’ contexts provide a distinct view of communities’ language from possible alternatives). That is, this paper could be even stronger if it showed more experiments that inform the reader about potential use cases where stance context representations are more favorable than other text-based representations.

I describe additional suggestions for improving this work in “Typos Grammar Style and Presentation Improvements” below, but did not include them in this section because I don’t think each point of negative-ish feedback is severe enough to be considered in paper decisions.


**Reproducibility:**

4: Could mostly reproduce the results, but there may be some variation because of sample variance or minor variations in their interpretation of the protocol or method.

**Reviewer Confidence:**

5: Positive that my evaluation is correct. I read the paper very carefully and I am very familiar with related work.

**Typos Grammar Style And Presentation Improvements:**

Line 86 says that no prior large-scale work has related variation in stance contexts to cross-platform engagement, which seems to imply that the current paper fills this gap by examining cross-platform engagement. However, the authors investigate cross-community engagement on a single platform (Reddit), so I would consider revising line 86 to change “cross-platform” to “cross-community” or something else.

Line 219: the authors state that they used Pavalanathan et al. 2017’s approach to extend the original stance intensifier lexicon to additional markers found on Reddit. I understand that due to page limit constraints, the authors redirect the reader to read the Appendix here, but it’d be helpful if the authors could also briefly describe the gist of the approach in the main text as well.

You should remove the keywords you list under the abstract as this is unusual for EMNLP.

Line 227-228 could be revised to clarify if the dataset contains the entire year of 2019 or just parts of 2019. Also, 2019 data is no longer considered “recent” in 2023, although I acknowledge that with the end of Pushshift earlier this year, obtaining more recent Reddit data is trickier now than before.

Lines 321-326 may be clearer as to why there are 10 features total by revising “…added an ‘extremeness’ version of each feature in our…” to “…added an ‘extremeness’ version of each feature introduced in \S 3.2.1 in our…”.

If the authors are looking to trim text for space reasons, I’d recommend shortening lines 357-358 into a single sentence as cosine similarity is a well-known approach in NLP venues and does not need much explanation.

---

> ### Author Rebuttal · Authors · 2023-08-28
>
> We greatly appreciate your constructive feedback, which will help us improve the paper. Since they are on related issues, we respond to your first reason to reject and first question for the authors at once.
>
> # First Reason to Reject + First Question for the Authors
>
> **Reason to Reject: Lines 210-224, which describe the process for deciding what stance intensifiers to use in this study, was confusing as to why the authors made the decisions that they did. The authors start with a wide pool of potential markers (89 intensifiers intersected with a list of 1000 stance markers) but then at the end they keep only 38. Why were so many intensifiers removed? The main text should be able to stand alone from the Appendix, but I also looked in the Appendix and it raised even more questions. For example, why use Bennett and Goodman (2018) to filter out which intensifiers to include?**
>
> **Question for the Authors: Since the paper frames itself as focusing on stancetaking, one may expect it to investigate a variety of stance markers. Would this study be extendable to other types of stance markers beyond intensifiers? Why did the authors restrict it to just intensifiers?**
>
> Two reviewers raised questions/concerns about related issues: why we focus on intensifiers, why we limited ourselves to just 38 intensifiers (and how those were selected), and whether the work generalizes to other intensifiers and other stance markers. We answer those questions here, with further detail on our motivations and methods, as well as additional experiments demonstrating generalizability; we’ll correspondingly address these issues in the camera-ready paper.
>
> Many reasons motivated the focus on intensifiers, some of which are summarized in the intro to Section 3 (which we will expand, and preview in the Intro), including their ubiquity, range of meaning, and variation associated with social factors. In addition, because stance markers cover a range of grammatical classes (verbs, adjectives, interjections, and others), we wanted to focus on a single class to avoid finding variation in stance contexts that is solely/largely due to variation in the grammatical contexts of such a diverse set of words. Future work will need to explore whether the contextual properties of other stance markers are equally informative about community behavior.
>
> We started with the intensifiers from Bennett & Goodman (2018) because they are known to communicate a range of degree of intensification (which was manually annotated in that work), and intersected this list with a list of stance markers likely to appear frequently on Reddit (generated using Pavalanathan et al.’s, 2017, methodology). We also removed two stopwords (*very* and *most*) given their pervasiveness in language, to ensure the words were informative.
>
> While the restriction to 38 words means we have to be cautious in assuming generalizability of all our results, this small set of words (chosen for task-independent reasons) is highly predictive of cross-posting behavior, indicating that stancetaking properties are indeed very powerful indicators of identity and community behavior.
>
> That said, we understand the concern that our results based on a small set of words may not generalize.  To address this, we performed the same cross-posting and social factors analyses (from Section 6) using the 252 intensifiers from Luo et al. (2019), on a set of 448 subreddits (selected for a different project).
>
> Across all new and submitted analyses, we achieve very consistent results using our stance context representations.  In predicting user cross-posting, stance context similarity significantly outperforms stance marker similarity in every case (Table 1 below).  The correlations between stance context distinctiveness and the 4 social factors also shows the exact same pattern as in the submitted results across all the new analyses, albeit with some reduction in power/magnitude of correlation when only using the 28 overlapping intensifiers (Table 2 below).
>
> Table 1: Results for our cross-posting analyses. We repeated the analyses on all 252 intensifiers in the 448 subreddits; we also did analyses on the 28 intensifiers that overlap between the Luo et al. dataset and our original 38 intensifiers so that we could compare the exact same set of intensifiers across the original 1798 subreddits and the 448 subreddits. The stance context representations significantly outperform the marker preference representations in all cases, at α=0.05 (using a corrected resampled t-test, as in the paper).
>
>
> $$
> \\begin{array}{l|r|r}
>  & \\text{Marker preference similarity} & \\text{Stance context similarity} \\\
> \text{Submitted Dataset (38 original markers)} & 75.1\\% & 82.3\\% \\\
> \text{Submitted Dataset (28 markers that overlap between datasets)} & 70.6\\% & 81.0\\% \\\
> \text{New Dataset (252 markers from Luo et al., 2019)} & 76.5\\% & 79.8\\% \\\
> \text{New Dataset  (28 markers that overlap between datasets)} & 73.3\\% & 78.5\\% \\\
> \\end{array}
> $$
>
>
> Table 2: Results for our social factors analyses. We repeated the analyses on all 252 intensifiers in the 448 subreddits; we also did analyses on the 28 intensifiers that overlap between the Luo et al. dataset and our original 38 intensifiers so that we could compare the exact same set of intensifiers across the original 1798 subreddits and the 448 subreddits. Bolded cells are significant at α=0.05, after applying Bonferroni correction.
> $$
> \\begin{array}{l|r|r|r|r}
> & \\text{Size} & \\text{Activity} & \\text{Loyalty} & \\text{Density} \\\
> \text{Submitted Dataset (38 original markers)} & \mathbf{0.21} & \mathbf{-0.36} & \mathbf{-0.34} & \mathbf{-0.34} \\\
> \text{Submitted Dataset (28 markers that overlap between datasets)} & \mathbf{0.18} & \mathbf{-0.36} & \mathbf{-0.26} & \mathbf{-0.30} \\\
> \text{New Dataset (252 markers from Luo et al., 2019)} & \mathbf{0.14} & \mathbf{-0.32} & \mathbf{-0.34} & \mathbf{-0.33} \\\
> \text{New Dataset  (28 markers that overlap between datasets)} & 0.13 & \mathbf{-0.35} & \mathbf{-0.29} & \mathbf{-0.31} \\\
> \\end{array}
> $$
>
> # Remaining Reasons to Reject
>
> **In the paper, the authors reduce Sentence BERT vectors into 10 features related to affect, formality, and politeness predicted from these vectors. I’m not fully convinced as to why politeness, affect (measured by NRC-VAD), and formality are key to measuring stancetaking. Why not just take the average of SBERT vectors instead of reducing those vectors down to these narrow linguistic properties? It’s possible that the 10-dimensional feature vectors offer interpretability, but it’s also possible that SBERT vectors may include richer information that would be helpful for understanding what concepts communities tend to take stances on.**
>
> The stance context properties we use have been theorized to be important aspects of how people position themselves during dialogue (Jaffe, 2009; Kiesling, 2009; Pavalanathan et al., 2017; Kiesling et al., 2018). SBERT representations almost certainly include additional semantic information, but this additional information may not be directly relevant to stancetaking. In addition, the interpretability offered by our methodology is an important aspect for us, as it allows us to better understand facets of community identity. For example, in addition to demonstrating that the Sports and Video Games subreddits are similar in stancetaking, we can see that a clear contributing factor is that both have distinctively high-arousal stance contexts, reflecting the competition-oriented nature of these subreddits. That said, an avenue for future work would be to do our analyses using SBERT representations to try to further determine the multitude of factors – stance context, vocabulary, topic, etc. – that relate to user dynamics on a social media platform.
>
> **When it comes to weighing the potential contribution or impact of this paper, I would emphasize Section 6 (relating stance to engagement behavior) more so than Section 5 (where the authors argue that intensifiers’ contexts provide a distinct view of communities’ language from possible alternatives). That is, this paper could be even stronger if it showed more experiments that inform the reader about potential use cases where stance context representations are more favorable than other text-based representations.**
>
> We felt that Section 5 was important to include as a basic check of soundness to illustrate that our stance context representations capture something different from other text/marker representations, and may shed light on relations among online communities. That being said, we agree with the reviewer that Section 6 is a more stringent analysis of our stance context representations, and those experiments are intended as our main contributions. While space prohibited further experimentation, in our final paper, we will include examples of other research questions that may be explored with our stance context representations.
>
> For example, in the paper we observed “mega” communities like Sports and Video Games that share stancetaking properties while having numerous fine-grained subcommunities. Further work with stance contexts may help shed additional light on why some communities are structured this way: in addition to establishing different subtopics (such as different sports teams), are “spin-off” communities (cf. Hessel et al., 2016) sometimes created so that users may discuss the same topic but taking an alternative stance? In addition, future work can consider how the stances that communities take may be important for understanding community growth and diversity. For example, on the male-dominated Reddit platform, does the use of more polite or higher valence stances lead to increased participation from women?
>
> # Remaining Questions for the Authors
>
> **Why is the text-based representation of communities left out of the user cross-posting prediction analysis (Table 2)?**
>
> Previous work has explored the efficacy of text-based measures on predicting cross-posting (Pavalanathan et al., 2017). In our work, we focus on the novel contribution of comparing how different aspects of stancetaking behaviour predict how users co-participate in online communities.
>
> # Missing References
>
> **Inducing Domain-Specific Sentiment Lexicons from Unlabeled Corpora, Hamilton et al. EMNLP 2016.**
>
> Thank you for pointing this out; we will discuss the Hamilton et al. (2016) paper in both our Related Work section and in the Limitations section (Section 8.2).
>
> # Typos Grammar Style And Presentation Improvements
>
> We appreciate the 6 editing suggestions below, and will implement each of them in our final paper.
>
> Regarding our dataset (point 4 below), we use the full year of data from 2019. Although (at the time we extracted it) the 2020 data was the most recent data available in the Reddit data dumps (Baumgartner et al., 2020), we opted for using 2019 data because we were concerned about the impact that Covid-19 and Reddit’s June 2020 ban of controversial communities may have had in that data. We will clarify this in the paper in Section 3.1.
>
> 1. **Line 86 says that no prior large-scale work has related variation in stance contexts to cross-platform engagement, which seems to imply that the current paper fills this gap by examining cross-platform engagement. However, the authors investigate cross-community engagement on a single platform (Reddit), so I would consider revising line 86 to change “cross-platform” to “cross-community” or something else.**
> 2. **Line 219: the authors state that they used Pavalanathan et al. 2017’s approach to extend the original stance intensifier lexicon to additional markers found on Reddit. I understand that due to page limit constraints, the authors redirect the reader to read the Appendix here, but it’d be helpful if the authors could also briefly describe the gist of the approach in the main text as well.**
> 3. **You should remove the keywords you list under the abstract as this is unusual for EMNLP.**
> 4. **Line 227-228 could be revised to clarify if the dataset contains the entire year of 2019 or just parts of 2019. Also, 2019 data is no longer considered “recent” in 2023, although I acknowledge that with the end of Pushshift earlier this year, obtaining more recent Reddit data is trickier now than before.**
> 5. **Lines 321-326 may be clearer as to why there are 10 features total by revising “…added an ‘extremeness’ version of each feature in our…” to “…added an ‘extremeness’ version of each feature introduced in \S 3.2.1 in our…”.**
> 6. **If the authors are looking to trim text for space reasons, I’d recommend shortening lines 357-358 into a single sentence as cosine similarity is a well-known approach in NLP venues and does not need much explanation.**
>
> # References
>
> Baumgartner, J., Zannettou, S., Keegan, B., Squire, M., & Blackburn, J. (2020, May). The pushshift reddit dataset. In *Proceedings of the international AAAI conference on web and social media* (Vol. 14, pp. 830-839).
>
> Bennett, E. D., & Goodman, N. D. (2018). Extremely costly intensifiers are stronger than quite costly ones. *Cognition, 178*, 147-161.
> Jaffe, A. (Ed.). (2009). *Stance: sociolinguistic perspectives*. Oup Usa.
>
> Kiesling, S. F. (2009). Style as stance. *Stance: sociolinguistic perspectives*, 171.
>
> Kiesling, S. F., Pavalanathan, U., Fitzpatrick, J., Han, X., & Eisenstein, J. (2018). Interactional stancetaking in online forums. *Computational Linguistics, 44*(4), 683-718.
>
> Pavalanathan, U., Fitzpatrick, J., Kiesling, S. F., & Eisenstein, J. (2017, July). A multidimensional lexicon for interpersonal stancetaking. In *Proceedings of the 55th Annual Meeting of the Association for Computational Linguistics (Volume 1: Long Papers)* (pp. 884-895).

---

### Official Review · Reviewer_NA9c · 2023-08-01

**Soundness:** 4

**Excitement:**

4: Strong: This paper deepens the understanding of some phenomenon or lowers the barriers to an existing research direction.

**Paper Topic And Main Contributions:**

This paper aims to model the sociolinguistic concept of stancetaking, in particular the "stance context", words immediately around stance markers. They aim to move beyond a focus on individual words/stance markers in computational modeling of stancetaking while also not requiring extensive manual annotation (as noted in lines 177-178). Focusing on intensifiers as stance markers, they extract features of affect (VAD), politeness, formality from contexts and use these as a representation of stance context, which they find contrasts subreddits in different ways than stance marker use and textual similarity. They then show that stance contexts can better predict cross-posting/community membership than just stance markers. Finally, they test if smaller, more active, denser communities have unique stancetaking. They find they do not, in contrast to prior work that shows that these types of communities are lexical innovators.

**Reasons To Accept:**

Stancetaking is an important and under-explored concept from sociolinguistics that can be useful in social NLP/computational social science. As the authors note, it has been shown to motivate many of the language choices users make and can act as an intermediary between linguistic features and identity. This paper does a great job motivating this need and bridging non-computational work in sociolinguistics with computational NLP work. It's an exemplary related work section, which is not an easy task given how unrelated these fields typically are.

Their qualitative analysis of differences in communities wrt stance context, stance markers, and textual similarity is informative, especially the examples of politics/education and sports/video games with low to moderate textual similarity but high stance context or stance marker similarity. The prediction task of cross-posting is convincing that stance context can capture individuality across subreddits, and the finding that similarity in formality and extremeness is most related to cross-posting behavior is valuable when thinking about what types of draw different online communities may have.

Their nuance to Lucy and Bamman 2021's findings that smaller, denser communities may have distinct vocabs but not distinct stancetaking is valuable to our knowledge of how language operates wrt identity in online communities.


**Reasons To Reject:**

The focus on intensifiers, and just 38 intensifiers in English, leads me to question some of the generalizability of the findings. Would these hold if other stance markers other than intensifiers were examined?

Some sort of embedding-based approach might be richer than SVD on tf-idf subreddit-word matrix for measuring the text similarity, but is probably not strictly necessary, especially if the goal is to match surface-level vocabulary instead of semantic vectors of words.

**Reproducibility:**

4: Could mostly reproduce the results, but there may be some variation because of sample variance or minor variations in their interpretation of the protocol or method.

**Reviewer Confidence:**

4: Quite sure. I tried to check the important points carefully. It's unlikely, though conceivable, that I missed something that should affect my ratings.

**Typos Grammar Style And Presentation Improvements:**

* It would provide a stronger motivation to give some example research questions that could be answered with stance contexts at the end of the intro. The second to last paragraph of the intro suggests some motivation for looking at stance contexts by connecting to affect, politeness, etc, but this is not super concrete on what can be done with it.
* More examples from the dataset of stances taken as captured in stance contexts would be illustrative
* The focus on intensifiers as stance marker of choice is quite important and should be mentioned earlier on.
* A clear definition of stance marker would be helpful in intro
* Line 378 , -> .

---

> ### Author Rebuttal · Authors · 2023-08-28
>
> We very much appreciate your constructive feedback, which will help us improve the paper, as detailed below.
>
> # Reasons to Reject
>
> **The focus on intensifiers, and just 38 intensifiers in English, leads me to question some of the generalizability of the findings. Would these hold if other stance markers other than intensifiers were examined?**
>
> Two reviewers raised questions/concerns about related issues: why we focus on intensifiers, why we limited ourselves to just 38 intensifiers (and how those were selected), and whether the work generalizes to other intensifiers and other stance markers. We answer those questions here, with further detail on our motivations and methods, as well as additional experiments demonstrating generalizability; we’ll correspondingly address these issues in the camera-ready paper.
>
> Many reasons motivated the focus on intensifiers, some of which are summarized in the intro to Section 3 (which we will expand, and preview in the Intro), including their ubiquity, range of meaning, and variation associated with social factors. In addition, because stance markers cover a range of grammatical classes (verbs, adjectives, interjections, and others), we wanted to focus on a single class to avoid finding variation in stance contexts that is solely/largely due to variation in the grammatical contexts of such a diverse set of words. Future work will need to explore whether the contextual properties of other stance markers are equally informative about community behavior.
>
> We started with the intensifiers from Bennett & Goodman (2018) because they are known to communicate a range of degree of intensification (which was manually annotated in that work), and intersected this list with a list of stance markers likely to appear frequently on Reddit (generated using Pavalanathan et al.’s, 2017, methodology). We also removed two stopwords (*very* and *most*) given their pervasiveness in language, to ensure the words were informative.
>
> While the restriction to 38 words means we have to be cautious in assuming generalizability of all our results, this small set of words (chosen for task-independent reasons) is highly predictive of cross-posting behavior, indicating that stancetaking properties are indeed very powerful indicators of identity and community behavior.
>
> That said, we understand the concern that our results based on a small set of words may not generalize.  To address this, we performed the same cross-posting and social factors analyses (from Section 6) using the 252 intensifiers from Luo et al. (2019), on a set of 448 subreddits (selected for a different project).
>
> Across all new and submitted analyses, we achieve very consistent results using our stance context representations.  In predicting user cross-posting, stance context similarity significantly outperforms stance marker similarity in every case (Table 1 below).  The correlations between stance context distinctiveness and the 4 social factors also shows the exact same pattern as in the submitted results across all the new analyses, albeit with some reduction in power/magnitude of correlation when only using the 28 overlapping intensifiers (Table 2 below).
>
> Table 1: Results for our cross-posting analyses. We repeated the analyses on all 252 intensifiers in the 448 subreddits; we also did analyses on the 28 intensifiers that overlap between the Luo et al. dataset and our original 38 intensifiers so that we could compare the exact same set of intensifiers across the original 1798 subreddits and the 448 subreddits. The stance context representations significantly outperform the marker preference representations in all cases, at α=0.05 (using a corrected resampled t-test, as in the paper).
>
>
> $$
> \\begin{array}{l|r|r}
>  & \\text{Marker preference similarity} & \\text{Stance context similarity} \\\
> \text{Submitted Dataset (38 original markers)} & 75.1\\% & 82.3\\% \\\
> \text{Submitted Dataset (28 markers that overlap between datasets)} & 70.6\\% & 81.0\\% \\\
> \text{New Dataset (252 markers from Luo et al., 2019)} & 76.5\\% & 79.8\\% \\\
> \text{New Dataset  (28 markers that overlap between datasets)} & 73.3\\% & 78.5\\% \\\
> \\end{array}
> $$
>
>
> Table 2: Results for our social factors analyses. We repeated the analyses on all 252 intensifiers in the 448 subreddits; we also did analyses on the 28 intensifiers that overlap between the Luo et al. dataset and our original 38 intensifiers so that we could compare the exact same set of intensifiers across the original 1798 subreddits and the 448 subreddits. Bolded cells are significant at α=0.05, after applying Bonferroni correction.
> $$
> \\begin{array}{l|r|r|r|r}
> & \\text{Size} & \\text{Activity} & \\text{Loyalty} & \\text{Density} \\\
> \text{Submitted Dataset (38 original markers)} & \mathbf{0.21} & \mathbf{-0.36} & \mathbf{-0.34} & \mathbf{-0.34} \\\
> \text{Submitted Dataset (28 markers that overlap between datasets)} & \mathbf{0.18} & \mathbf{-0.36} & \mathbf{-0.26} & \mathbf{-0.30} \\\
> \text{New Dataset (252 markers from Luo et al., 2019)} & \mathbf{0.14} & \mathbf{-0.32} & \mathbf{-0.34} & \mathbf{-0.33} \\\
> \text{New Dataset  (28 markers that overlap between datasets)} & 0.13 & \mathbf{-0.35} & \mathbf{-0.29} & \mathbf{-0.31} \\\
> \\end{array}
> $$
>
>
> **Some sort of embedding-based approach might be richer than SVD on tf-idf subreddit-word matrix for measuring the text similarity, but is probably not strictly necessary, especially if the goal is to match surface-level vocabulary instead of semantic vectors of words.**
>
> Your observation about matching surface-level vocabulary is indeed along the lines of our thinking. The very richness of embeddings means that they are likely to capture some of the aspects of stance that we’re trying to contrast with more direct textual similarity. Our use of SVD follows from similar work investigating the relationship between community textual similarity and and other targeted aspects of community language (Lucy & Mendelsohn, 2019; Pavalanathan et al., 2017).
>
> # Typos Grammar Style And Presentation Improvements
>
> **It would provide a stronger motivation to give some example research questions that could be answered with stance contexts at the end of the intro. The second to last paragraph of the intro suggests some motivation for looking at stance contexts by connecting to affect, politeness, etc, but this is not super concrete on what can be done with it.**
>
> In our final paper, we will be clearer about the research questions that our stance context representations can help answer. For example, while we mention user engagement and network properties in the intro, it may not be clear that our findings reveal interesting aspects of the structure of Reddit: “mega” communities like Sports and Video Games that share stancetaking properties while having numerous fine-grained subcommunities. Further work with stance contexts may help shed additional light on why some communities are structured this way: in addition to establishing different subtopics (such as different sports teams), are “spin-off” communities (cf. Hessel et al., 2016) sometimes created so that users may discuss the same topic but taking an alternative stance? In addition, future work can consider how the stances that communities take may be important for understanding community growth and diversity. For example, on the male-dominated Reddit platform, does the use of more polite or higher valence stances lead to increased participation from women?
>
> **More examples from the dataset of stances taken as captured in stance contexts would be illustrative**
>
> We can include example stance context representations for sentences from our dataset in the final paper.  To give a sense of the data here, we provide the feature values for two of the examples in the intro that differ in valence, as well as another sentence that differs in formality:
>
> $$
> \\begin{array}{l|r|r|r|r|r}
> & \\text{Valence} & \\text{Arousal} & \\text{Dominance} & \\text{Politeness} & \\text{Formality} \\\
> \text{I hope you are incredibly proud of yourself.} & 0.77 & 0.67 & 0.57 & 0.70 & 0.32 \\\
> \text{Both of you are incredibly stupid} & 0.16 & 0.61 & 0.27 & 0.25 & 0.22 \\\
> \text{This would be incredibly valuable scientifically for studying the origins of life.} & 0.65 & 0.58 & 0.74 & 0.64 & 0.91
> \\end{array}
> $$
>
>
>
>
> **The focus on intensifiers as stance marker of choice is quite important and should be mentioned earlier on.**
>
> We agree, and will indicate our focus on intensifiers and motivation for doing so in the introduction rather than waiting until Section 3.
>
> **A clear definition of stance marker would be helpful in intro**
>
> In the final paper, we will expand on the definition of stance markers as follows: “Stance markers are words that demarcate stance, including, among others, intensifiers (*really*, *insanely*, *terribly*), modals (*might*, *should*), and evaluative words (*like*, *love*, *hate*)”.
>
> **Line 378 , -> .**
>
> We will fix the typo on line 378.
>
> # References
>
> Bennett, E. D., & Goodman, N. D. (2018). Extremely costly intensifiers are stronger than quite costly ones. *Cognition*, 178, 147-161.
>
> Hessel, J., Tan, C., & Lee, L. (2016). Science, askscience, and badscience: On the coexistence of highly related communities. In *Proceedings of the international AAAI conference on web and social media* (Vol. 10, No. 1, pp. 171-180).
>
> Luo, Y., Jurafsky, D., & Levin, B. (2019, August). From insanely jealous to insanely delicious: Computational models for the semantic bleaching of English intensifiers. In *Proceedings of the 1st International Workshop on Computational Approaches to Historical Language Change* (pp. 1-13).
>
> Pavalanathan, U., Fitzpatrick, J., Kiesling, S. F., & Eisenstein, J. (2017, July). A multidimensional lexicon for interpersonal stancetaking. In *Proceedings of the 55th Annual Meeting of the Association for Computational Linguistics (Volume 1: Long Papers)* (pp. 884-895).

---

### Official Review · Reviewer_Qr3C · 2023-08-05

**Soundness:** 3

**Excitement:**

3: Ambivalent: It has merits (e.g., it reports state-of-the-art results, the idea is nice), but there are key weaknesses (e.g., it describes incremental work), and it can significantly benefit from another round of revision. However, I won't object to accepting it if my co-reviewers champion it.

**Paper Topic And Main Contributions:**

This paper proposes a stance context representation that can capture community identity patterns differently from textual or stance marker measures. The properties of the representation consist from the 5 features of affect, politeness and formality. For each feature, the paper prepares a model based on SBERT to assign the feature score to a sentence. The representation is evaluated on a Reddit dataset with inter- and intra-community patterns and the evaluation highlights the strength of using the proposed context representation.

**Questions For The Authors:**

1. Can you measure the human correlations of the model-based scores presented in Section 3.2.1? There may not be a dataset annotated with the proposed features but I believe the construction of a small evaluation dataset is possible with some annotator costs.
2. Why have you chose SBERT instead of BERT in the models proposed in Section 3.2.1? Since these models are regression models, I believe the use of CLS token in BERT is simpler than the use toke-level mean-pooling representation of SBERT.
3. Could you elaborate more how the three topic-topic pairs are chosen in Section 5.2? Are these pairs randomly chosen or are these high similarity pairs for Text, Stance and Marker? The characteristic found in Sports & Video Games pair is interesting but I wonder whether this kind of pair is common in Reddit.
4. Can't you examine the Spearman correlations of the social factors against the Textual similarity in Section 6.1? Since Textual vs. Stance Context has a high correlation in Table 1, I wonder the Textual similarity can also capture the characteristics seen in the Stance Context similarity.

**Reasons To Accept:**

1. The proposed representation has shown that it can capture a topic-topic similarity such as one between Sports & Video Games which can not be seen in a textual similarity or a marker similarity.
2. It is interesting to see that the social factors of the cross-posting behavior is quite different in the stance context and the marker preference.

**Reasons To Reject:**

1. The proposed stance context features are model-based but their qualities are not confirmed against ground truth scores such as human scores.
2. The comparison of topic-topic similarities and the investigation of the cross-posting behavior are quite limited. I will further elaborate this weakness in Q3 and Q4 in the following questions section.

**Reproducibility:**

3: Could reproduce the results with some difficulty. The settings of parameters are underspecified or subjectively determined; the training/evaluation data are not widely available.

**Reviewer Confidence:**

2: Willing to defend my evaluation, but it is fairly likely that I missed some details, didn't understand some central points, or can't be sure about the novelty of the work.

---

> ### Author Rebuttal · Authors · 2023-08-28
>
> We very much appreciate your constructive feedback, which will help us improve the paper, as detailed below. We respond to your first reason to reject when responding to your first question, and your second reason to reject when responding to your third and fourth questions.
>
> # Questions for the Authors
> **First reason to reject: The proposed stance context features are model-based but their qualities are not confirmed against ground truth scores such as human scores.**
>
> **Q1: Can you measure the human correlations of the model-based scores presented in Section 3.2.1? There may not be a dataset annotated with the proposed features but I believe the construction of a small evaluation dataset is possible with some annotator costs.**
>
> We tested our feature models in Section 3.2.1 on held-out portions of datasets with human annotations of the features, and all models achieved accuracies comparable to prior work. (The models achieved accuracies of .85, .77., .80, and .76 for valence, arousal, dominance, and formality in comparison to the gold standard; for politeness, we achieved in-domain accuracy of .84 [cross-validation on Wikipedia] and cross-domain of .65 [train Wikipedia/test Stack Exchange].) This was reported in Appendix C and will be summarized in the main body of the camera-ready paper.
>
>
> **Q2: Why have you chose SBERT instead of BERT in the models proposed in Section 3.2.1? Since these models are regression models, I believe the use of CLS token in BERT is simpler than the use toke-level mean-pooling representation of SBERT.**
>
> We used SBERT because it has been shown to outperform the BERT CLS token on three sentiment prediction tasks (Reimers & Gurevych, 2019). Since our stance context features are high-level text properties similar to sentiment, we expected SBERT representations to be better for predicting our stance context features. We will make this motivation more explicit in Section 3.2 in the camera ready.
>
> **Second reason to reject: The comparison of topic-topic similarities and the investigation of the cross-posting behavior are quite limited. I will further elaborate this weakness in Q3 and Q4 in the following questions section.**
>
> **Q3: Could you elaborate more how the three topic-topic pairs are chosen in Section 5.2? Are these pairs randomly chosen or are these high similarity pairs for Text, Stance and Marker? The characteristic found in Sports & Video Games pair is interesting but I wonder whether this kind of pair is common in Reddit.**
>
> We manually selected the pairs in Section 5.2 as illustrations of differences in what each of our three representations capture. Some other pairs of subreddit topics also pattern like Sports and Video Games –  i.e., having much higher stance context similarity than text similarity; e.g., Sports and Technology, NSFW and Politics, and Locations and Entertainment. For the camera-ready version we are exploring visualizations that would make it easier to identify pairs of subreddits that differ in their similarity according to different representations.
>
>
>
> **Q4: Can't you examine the Spearman correlations of the social factors against the Textual similarity in Section 6.1? Since Textual vs. Stance Context has a high correlation in Table 1, I wonder the Textual similarity can also capture the characteristics seen in the Stance Context similarity.**
>
> Lucy and Bamman (2021) have shown how the social factors relate to distinctiveness in vocabulary (which we replicate on our data in Appendix G). We wanted to build on this work by exploring the role of contextual properties beyond individual words and word senses, hence the focus on our novel contribution of how the social factors relate to distinctiveness in aspects of stancetaking. We will clarify this in Section 6 of the camera-ready paper.
>
>
> # References
> Reimers, N., & Gurevych, I. (2019, November). Sentence-BERT: Sentence Embeddings using Siamese BERT-Networks. In *Proceedings of the 2019 Conference on Empirical Methods in Natural Language Processing and the 9th International Joint Conference on Natural Language Processing (EMNLP-IJCNLP)* (pp. 3982-3992).
>
> Lucy, L., & Bamman, D. (2021). Characterizing English variation across social media communities with BERT. *Transactions of the Association for Computational Linguistics*, 9, 538-556.

---

### Meta-Review · Area_Chair_GBgS · 2023-09-14

**Recommendation:** 4

**Metareview:**

On the whole, reviewers thought that this was an interesting paper. They appreciated the sociolinguistic motivations for the focus/approach, and the ways that the proposed context representations could enrich our understanding of communities, beyond previous literature.

The reviewers had a few concerns about the methodological choices and paper framing. They felt that these concerns could be addressed via some writing changes, especially since the authors started to provided clarifications in their response. I’ve highlighted some main suggestions, and I encourage the authors to attend to these suggestions — as well as others that the reviewers raised — as they revise their paper.
* Reviewers found the choice of focusing on intensifiers somewhat arbitrary. It would be great if they could elaborate on/foreground the point briefly made, that controlling for grammatical variation is important, and if they could include the analysis with the full set of markers presently included in their response to reviewers. In particular, reviewers feel that including the additional analyses would greatly improve the soundness of the paper.
* Reviewers note that the additional explanatory power offered by the proposed representations is a strength of the paper, and ought to be emphasized. For instance, one reviewer suggests emphasizing S6, relating stance to behavior, over S5 — in the interests of showing the reader examples of _how_ the representations could be used in other analyses.
* More work could be done to validate the representations, especially with human judgements.

---

### Decision · Program_Chairs · 2023-10-07

**Decision:**

Accept-Findings

**Comment:**

On the whole, reviewers thought that this was an interesting paper. They appreciated the sociolinguistic motivations for the focus/approach, and the ways that the proposed context representations could enrich our understanding of communities, beyond previous literature.

The reviewers had a few concerns about the methodological choices and paper framing. They felt that these concerns could be addressed via some writing changes, especially since the authors started to provided clarifications in their response. I’ve highlighted some main suggestions, and I encourage the authors to attend to these suggestions — as well as others that the reviewers raised — as they revise their paper.
* Reviewers found the choice of focusing on intensifiers somewhat arbitrary. It would be great if they could elaborate on/foreground the point briefly made, that controlling for grammatical variation is important, and if they could include the analysis with the full set of markers presently included in their response to reviewers. In particular, reviewers feel that including the additional analyses would greatly improve the soundness of the paper.
* Reviewers note that the additional explanatory power offered by the proposed representations is a strength of the paper, and ought to be emphasized. For instance, one reviewer suggests emphasizing S6, relating stance to behavior, over S5 — in the interests of showing the reader examples of _how_ the representations could be used in other analyses.
* More work could be done to validate the representations, especially with human judgements.